# PROVABLY CONTINUAL UNLEARNING FOR LARGE LANGUAGE MODEL

## ABSTRACT

Continual unlearning in large language models (LLMs) requires forgetting targeted domains while preserving utility elsewhere as requests arrive sequentially. Existing approaches are largely heuristic and accumulate interference over time. We present a principled *optimization* framework **SCOPE** (**S**pectral **O**rthogonality for **C**ontinual unlearning with **P**rovable guarant**E**es) that formalizes continual unlearning via three explicit conditions: *selective forgetting*, *utility preservation*, and *persistence*, and satisfies them by parameterizing updates in an orthonormal spectral basis with disjoint coefficient supports. This construction enforces orthogonality by design, yields capacity laws that bound interference as requests accumulate, and admits an efficient FFT-based instantiation that needs no basis storage and scales as $O(d \log d)$. The same parameterization provides an inference-time routing signal via spectral activations, enabling calibrated triggering of unlearning adapters. Across discriminative, generative, and reasoning benchmarks-and without using retained data from unaffected domains where our method delivers stronger unlearning–utility trade-offs and more stable scaling than competitive baselines, offering a scalable framework with explicit guarantees for continual unlearning in LLMs.

## 1 INTRODUCTION

Large language models (LLMs) have revolutionized natural language processing through their ability to encode vast amounts of knowledge from diverse domains (Wang et al., 2024). However, this capability raises critical concerns about privacy, copyright, and safety (Pan et al., 2020; Liu et al., 2024). Deployed LLMs frequently encounter scenarios where specific information must be removed on demand. Examples include: (1) toxic or offensive content flagged by users, (2) copyrighted passages targeted by legal requests, and (3) outdated or misleading facts as knowledge evolves. The task of *machine unlearning* (Bourtoule et al., 2021) addresses this challenge by enabling the selective removal of unwanted data influences while preserving model utility on all remaining domains.

Most existing work considers unlearning as a single-shot operation, where the model is asked to forget one domain or dataset. In practice, however, unlearning requests are not one-time events but arrive *sequentially* over time. A realistic deployment must support multiple, evolving requests: after forgetting a toxic subset, a model may later need to remove copyrighted material, and later still discard outdated knowledge. This *continual unlearning* setting is more challenging than isolated unlearning because each update must satisfy three conditions: *selective forgetting* (the current request is forgotten), *utility preservation* (performance on other domains is maintained), and *persistence* (previously forgotten domains remain forgotten after later updates). Meeting all three together is nontrivial, as each operation shrinks the feasible parameter space.

Current LLM unlearning methods can be broadly categorized into two paradigms: **parameter optimization methods** and **in-context prompting methods**. Parameter optimization modifies model weights directly, typically by applying gradient ascent on unlearning data, optimizing with shuffled or rejection labels, or restricting updates to selected parameter subsets (Chen & Yang, 2023; Eldan & Russinovich, 2023; Jia et al., 2024; Zhang et al., 2024). In-context prompting instead alters model behavior by modifying prompts to elicit refusal responses (Thaker et al., 2024; Pawelczyk et al., 2024). While both strategies can achieve forgetting in isolated cases, they suffer from serious limitations in the continual setting. Prompt-based methods generally preserve utility but do not truly erase knowledge from the model parameters, making persistence unreliable. Parameter optimization

can enforce forgetting more strongly but relies on heuristics such as surrogate losses or orthogonal regularizers. These heuristics lack theoretical guarantees, and when applied sequentially, they cause unpredictable interference between tasks and cumulative degradation of utility (Gu et al., 2024; Gupta et al., 2024). Recent work (Gao et al., 2025) attempts to address these issues using orthogonality constraints on LoRA adapters (Hu et al., 2021), but this approach still suffers from two drawbacks: (i) optimization may converge to local minima with substantial interference, and (ii) no explicit bounds exist on how performance degrades as unlearning requests accumulate.

We address these limitations by introducing the first *optimization-based framework with explicit guarantees* for continual unlearning in LLMs. We propose the principled *optimization* framework **SCOPE** (**S**pectral **O**rthogonality for **C**ontinual unlearning with **P**rovable guarant**E**es). Our approach begins by formalizing the problem through three explicit conditions: selective forgetting, utility preservation, and persistence. We then analyze the constrained optimization problem defined by these conditions and show updates must be restricted to structured subspaces to satisfy reliably. This analysis naturally motivates a spectral decomposition of parameter updates, where weight updates are expressed in an orthonormal basis and assigned disjoint coefficient supports across tasks. This construction enforces orthogonality *by design*, ensuring utility preservation and persistence constraints are automatically satisfied, while also yielding capacity laws quantifying the number of requests that can be handled without interference. Unlike heuristic orthogonalization strategies, our method provides provable guarantees within capacity and controlled degradation beyond it. Although the spectral framework is basis-agnostic, we instantiate it with Fast Fourier Transform (FFT) bases in practice. This choice offers three advantages: (i) no explicit basis storage is required, (ii) updates can be computed in $O(d \log d)$ time using efficient implementations such as `torch.fft`, and (iii) Fourier bases empirically capture compact energy in transformer layers. Furthermore, the spectral parameterization yields an inference-time signal: spectral activation norms correlate with domain relevance, enabling the model to automatically route inputs to appropriate unlearning adapters. Thus, our framework unifies optimization-time updates with inference-time routing, eliminating need for auxiliary out-of-distribution detectors. In summary, our work makes the following contributions:

- We establish a principled optimization framework for continual unlearning, explicitly formalizing the three fundamental conditions of forgetting, preservation, and persistence, and systematically deriving update constraints that make them practically achievable.

- We propose a novel spectral decomposition method that enforces these conditions strictly by construction, admits an efficient FFT-based instantiation, and provides explicit theoretical capacity laws with provable interference bounds.

- We unify parameter optimization and inference-time routing within the same framework, enabling scalable deployment without auxiliary detection modules.

Extensive experiments across discriminative, generative, and reasoning benchmarks show that our approach consistently outperforms state-of-the-art baselines in unlearning–utility trade-offs, scales to long sequences of requests, and avoids the cumulative degradation observed in existing methods.

## 2 RELATED WORK

**Machine Unlearning for Large Language Models.** Machine unlearning (Bourtoule et al., 2021) addresses the "right to be forgotten" (GDPR, 2018; Pardau, 2018) by enabling the removal of specific data influences from trained models. Early approaches relied on exact unlearning through retraining from scratch, which is computationally prohibitive for large-scale LLMs (Wang et al., 2024). This motivated the development of approximate unlearning methods that aim to remove the effect of target data with significantly reduced overhead. Current LLM unlearning techniques can be broadly divided into two paradigms: **parameter optimization methods** and **in-context learning methods**. Parameter optimization approaches (Chen & Yang, 2023; Eldan & Russinovich, 2023; Jia et al., 2024; Zhang et al., 2024; Meng et al., 2022; Li et al., 2024) directly modify model weights through strategies such as gradient ascent on unlearning data (Golatkar et al., 2020; Yao et al., 2023), preference optimization with shuffled or rejection labels (Eldan & Russinovich, 2023; Zhang et al., 2024), or localizing parameters and updating only selected subsets (Yu et al., 2023). In-context learning methods (Thaker et al., 2024; Pawelczyk et al., 2024), by contrast, modify prompts to elicit refusal responses for undesired content without changing the underlying parameters. While parameter

optimization methods typically achieve stronger forgetting performance than in-context methods, they generally assume access to substantial retained datasets to preserve model utility. This assumption is increasingly unrealistic for LLMs trained on massive, proprietary corpora (Liu et al., 2024; Sun et al., 2024). Moreover, existing approaches largely treat unlearning as isolated, single-task operations and neglect the continual nature of real-world unlearning requests.

**Continual Learning and Catastrophic Forgetting.** The continual unlearning challenge is closely related to catastrophic forgetting in continual learning (Gu et al., 2024; Gupta et al., 2024). However, while continual learning seeks to preserve knowledge across tasks, continual unlearning requires the opposite: selectively forgetting specific knowledge while maintaining everything else. This difference makes continual unlearning even more challenging, as it combines the need for targeted degradation with strong utility preservation. Recent studies have begun to examine sequential unlearning scenarios (Gu et al., 2024), showing that naive reuse of single-task methods leads to cumulative degradation of utility across unaffected domains. These works, however, remain primarily empirical and lack theoretical grounding. They do not provide a principled framework for analyzing or mitigating interference between sequential unlearning operations, leaving open the question of how to design methods with provable guarantees in the continual setting. Most relevant to our setting is the $O^3$ framework (Gao et al., 2025), which introduces orthogonal LoRA adapters together with a glocal-aware OOD detector. $O^3$ demonstrates strong empirical performance across multiple benchmarks without requiring retained data, showing that explicit mechanisms for disentangling task updates and routing can substantially mitigate cumulative degradation. Nevertheless, $O^3$ relies on heuristic orthogonalization and does not provide provable guarantees on interference or capacity, leaving open the question of how to design constructive methods with explicit bounds.

Unlike continual learning, where stability and plasticity trade-offs have been formalized, continual unlearning has not yet been framed in terms of explicit conditions that algorithms must satisfy. This missing formulation is precisely what we establish in Section 2.

# 3 PROBLEM FORMULATION

We now establish the optimization foundation for continual unlearning in large language models (LLMs). Our overarching goal is to precisely and systematically define the sequential unlearning problem, introduce the three explicit conditions: *selective forgetting*, *utility preservation*, and *persistence* – and clearly show how these conditions naturally lead to a principled constrained optimization view. This theoretical analysis motivates the spectral solution framework developed in Section 4.

## 3.1 PROBLEM DEFINITION

Consider an LLM $M_{\boldsymbol{\theta}}$ with parameters $\boldsymbol{\theta} \in \mathbb{R}^d$. Our formulation is entirely agnostic to the specific model architecture and underlying pretraining objective.

**Domain structure.** We assume model knowledge decomposes into $K$ domains indexed by $i \in \{1, \ldots, K\}$, each associated with distribution $\mathcal{P}_i$ over input–output $(\boldsymbol{x}, \boldsymbol{y})$. Define expected loss

$$L_i(\boldsymbol{\theta}) = \mathbb{E}_{(\boldsymbol{x}, \boldsymbol{y}) \sim \mathcal{P}_i}\big[\ell(M_{\boldsymbol{\theta}}(\boldsymbol{x}), \boldsymbol{y})\big], \tag{1}$$

where $\ell$ is the prediction loss. In practice, domain representations overlap, so forgetting is not a simple masking operation but requires careful optimization.

**Sequential requests.** We consider a sequence of unlearning requests

$$\mathcal{R} = \{(j_1, \mathcal{D}_1^U), \ldots, (j_T, \mathcal{D}_T^U)\},$$

where $j_t$ denotes the target domain at step $t$ and $\mathcal{D}_t^U$ is a dataset drawn from $\mathcal{P}_{j_t}$. Let $\boldsymbol{\theta}_t$ be the parameters after processing request $t$, with $\boldsymbol{\theta}_0$ the pretrained initialization. We assume no retained data from non-target domains is available for optimization-only the forget sets $\{\mathcal{D}_t^U\}$ are provided.

**Definition 3.1** (Continual unlearning). *An algorithm maps $\boldsymbol{\theta}_0 \mapsto \{\boldsymbol{\theta}_1, \ldots, \boldsymbol{\theta}_T\}$ such that the final model $M_{\boldsymbol{\theta}_T}$ satisfies:*

    *1. **Selective forgetting:** performance on each requested domain degrades measurably;*

2. ***Utility preservation:*** *performance on all non-requested domains remains within tolerance;*

3. **Persistence:** *previously forgotten domains remain degraded after subsequent updates.*

## 3.2 OPTIMIZATION VIEW

**Vectorization and gradients.** Parameters are partitioned as $\{\boldsymbol{W}^{(\ell)}\}_{\ell=1}^{L}$; updates as $\Delta\boldsymbol{\theta} = \mathrm{vec}(\{\Delta\boldsymbol{W}^{(\ell)}\})$. Gradients decompose similarly: $g_i(\boldsymbol{\theta}) = \mathrm{vec}(\{\boldsymbol{G}_i^{(\ell)}\})$, with $\boldsymbol{G}_i^{(\ell)} = \nabla_{\boldsymbol{W}^{(\ell)}} L_i(\boldsymbol{\theta})$. We use Frobenius inner products $\langle g_i(\boldsymbol{\theta}), \Delta\boldsymbol{\theta}\rangle = \sum_{\ell=1}^{L}\langle \boldsymbol{G}_i^{(\ell)}, \Delta\boldsymbol{W}^{(\ell)}\rangle_F$ and $\langle \boldsymbol{A}, \boldsymbol{B}\rangle_F = \mathrm{tr}(\boldsymbol{A}^\top \boldsymbol{B})$.

**First-order approximation.** For sufficiently small updates,
$$L_i(\boldsymbol{\theta}_t) \approx L_i(\boldsymbol{\theta}_{t-1}) + \langle g_i(\boldsymbol{\theta}_{t-1}), \Delta\boldsymbol{\theta}_t\rangle, \tag{2}$$
with higher-order terms absorbed into tolerances defined below. This follows from standard smoothness assumptions: if $L_i$ has $L$-Lipschitz gradients, the error is $O(\|\Delta\boldsymbol{\theta}_t\|^2)$.

**Constrained optimization.** Ideally, request $t$ maximizes forgetting on domain $j_t$ while leaving others unaffected:
$$\max_{\Delta\boldsymbol{\theta}_t} \langle g_{j_t}(\boldsymbol{\theta}_{t-1}), \Delta\boldsymbol{\theta}_t\rangle \quad \text{s.t.} \quad \langle g_i(\boldsymbol{\theta}_{t-1}), \Delta\boldsymbol{\theta}_t\rangle = 0, \ \forall i \in \mathcal{P}_t, \tag{3}$$
where $\mathcal{P}_t = \{1, \ldots, K\} \setminus \{j_t\}$ includes both retained and previously forgotten domains. As $t$ grows, the feasible subspace shrinks, which explains the difficulty of continual unlearning.

## 3.3 EXPLICIT CONDITIONS

The optimization constraints can be relaxed into measurable conditions.

**Definition 3.2** (Selective forgetting). *For target $j_t$,*
$$L_{j_t}(\boldsymbol{\theta}_t) \geq L_{j_t}(\boldsymbol{\theta}_{t-1}) + \epsilon_t, \quad \epsilon_t > 0.$$

**Definition 3.3** (Utility preservation). *For all $i \notin \{j_1, \ldots, j_t\}$,*
$$|L_i(\boldsymbol{\theta}_t) - L_i(\boldsymbol{\theta}_{t-1})| \leq \delta.$$

**Definition 3.4** (Persistence). *For $s < t$,*
$$L_{j_s}(\boldsymbol{\theta}_t) \geq L_{j_s}(\boldsymbol{\theta}_s) - \eta.$$

$\epsilon_t$ enforces meaningful forgetting for the current unlearning request, $\delta$ bounds allowable drift on preserved domains, and $\eta$ prevents inadvertent relearning of previously forgotten domains. These tolerances absorb higher-order errors and stochasticity.

**Proposition 3.5** (Gradient constraint condition). *Under first-order approximation, the three conditions imply:*
$$\textit{Forgetting: } \langle g_{j_t}(\boldsymbol{\theta}_{t-1}), \Delta\boldsymbol{\theta}_t\rangle \geq \epsilon_t, \tag{4}$$
$$\textit{Preservation: } |\langle g_i(\boldsymbol{\theta}_{t-1}), \Delta\boldsymbol{\theta}_t\rangle| \leq \delta, \quad i \notin \{j_1, \ldots, j_t\}, \tag{5}$$
$$\textit{Persistence: } |\langle g_{j_s}(\boldsymbol{\theta}_{t-1}), \Delta\boldsymbol{\theta}_t\rangle| \leq \eta, \quad s < t. \tag{6}$$

As $t$ increases, the number of constraints rapidly grows and naive optimization soon becomes infeasible. This strongly motivates structured parameterizations that reliably satisfy orthogonality by construction rather than by penalty. In Section 4, we introduce a principled spectral decomposition where coefficient supports can be flexibly allocated across tasks, thereby enabling exact satisfaction of the three conditions within capacity and quantifiable degradation beyond it.

## 4 SPECTRAL SOLUTION FRAMEWORK

We now develop a spectral decomposition approach that provides provable guarantees for continual unlearning. The key idea is to parameterize updates in a basis where orthogonality is enforced by construction, so that the preservation and persistence constraints from Section 2 are automatically satisfied. This yields the first constructive framework that meets all three conditions of continual unlearning, provides explicit capacity laws, and extends naturally to inference-time routing.

## 4.1 SPECTRAL PARAMETERIZATION

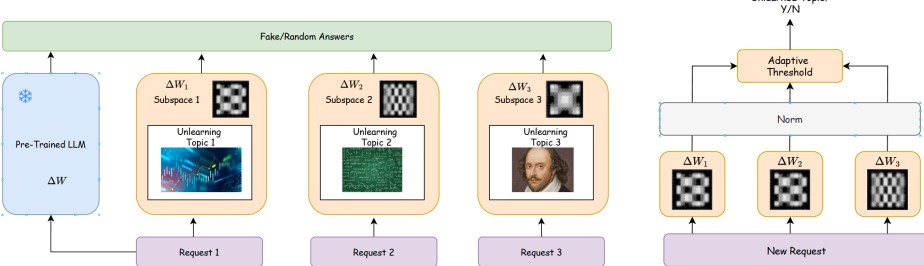

(a) Spectral solution framework        (b) Spectral activation norm

Figure 1: Spectral solution framework: The natural Orthogonality between adapters (form by FFT) for unlearning requested knowledge (Fig. 1a) and Spectral activation norm (SAN) is used to detect whether the input contains the unlearning knowledge(Fig. 1b)

For each layer $\ell$, let $\boldsymbol{W}^{(\ell)} \in \mathbb{R}^{m_\ell \times n_\ell}$ be the weight matrix with update $\Delta \boldsymbol{W}^{(\ell)}$. We introduce orthonormal matrices $\boldsymbol{U}_L^{(\ell)} \in \mathbb{R}^{m_\ell \times k_\ell}$ and $\boldsymbol{U}_R^{(\ell)} \in \mathbb{R}^{n_\ell \times k_\ell}$, and parameterize updates as

$$\Delta \boldsymbol{W}_t^{(\ell)} = \boldsymbol{U}_L^{(\ell)} \boldsymbol{S}_t^{(\ell)} (\boldsymbol{U}_R^{(\ell)})^\top, \quad \boldsymbol{S}_t^{(\ell)} \in \mathbb{R}^{k_\ell \times k_\ell}.$$

Each $\boldsymbol{S}_t^{(\ell)}$ is sparse, with support set $\Omega_t^{(\ell)} \subseteq [k_\ell] \times [k_\ell]$. Define sparsity $\rho_\ell = |\Omega_t^{(\ell)}|/k_\ell^2$ and total budget $\mathcal{K} = \sum_{\ell=1}^L k_\ell^2$. Denote by $P_\Omega^{(\ell)}$ the projector onto entries in $\Omega^{(\ell)}$.

**Orthogonality by construction.** Inner products between gradients and updates factor through the spectral basis:

$$\langle \boldsymbol{G}_i^{(\ell)}, \Delta \boldsymbol{W}_t^{(\ell)} \rangle_F = \langle (\boldsymbol{U}_L^{(\ell)})^\top \boldsymbol{G}_i^{(\ell)} \boldsymbol{U}_R^{(\ell)}, \boldsymbol{S}_t^{(\ell)} \rangle_F.$$

Thus, if supports are disjoint across tasks, their contributions are orthogonal and interference vanishes.

**Theorem 4.1** (Automatic constraint satisfaction). *If $\Omega_t^{(\ell)} \cap \Omega_q^{(\ell)} = \emptyset$ for all $\ell$ and for all preserved or previously forgotten domains q, then $\langle g_q(\boldsymbol{\theta}_{t-1}), \Delta \boldsymbol{\theta}_t \rangle = 0$. Hence preservation and persistence are satisfied exactly, while forgetting is achieved whenever $\|P_{\Omega_t^{(\ell)}}((\boldsymbol{U}_L^{(\ell)})^\top \boldsymbol{G}_{j_t}^{(\ell)} \boldsymbol{U}_R^{(\ell)})\|_F > 0$ for some $\ell$.*

**FFT as a practical instantiation.** The theory applies to any orthonormal basis $\{\boldsymbol{U}_L^{(\ell)}, \boldsymbol{U}_R^{(\ell)}\}$. In practice, we use Fourier bases: (i) no need to store $\boldsymbol{U}$, (ii) efficient $O(n \log n)$ transforms via `torch.fft`, and (iii) empirical energy compaction in transformer layers. FFT thus offers an efficient and memory-free instantiation, though the guarantees are basis-agnostic.

**Complexity and Practicality.** For each layer $\ell$, computing $(\boldsymbol{U}_L^{(\ell)})^\top \boldsymbol{G}^{(\ell)} \boldsymbol{U}_R^{(\ell)}$ via FFT costs $O(m_\ell \log m_\ell + n_\ell \log n_\ell)$, plus $O(|\Omega_t^{(\ell)}|)$ sparse multiplications. Total memory scales as $O(\sum_{t,\ell} |\Omega_t^{(\ell)}|)$. Compared to direct constrained optimization (cubic in $d$), this approach scales as $O(d \log d)$ and requires no explicit storage of $\boldsymbol{U}$.

## 4.2 CAPACITY AND ALLOCATION

Each task consumes $\sum_\ell |\Omega_t^{(\ell)}| = \sum_\ell \rho_\ell k_\ell^2$ coefficients. Perfect isolation is possible until the global budget $\mathcal{K}$ is exhausted.

**Theorem 4.2** (Capacity bound). *Let $\bar{\rho} = \frac{1}{\mathcal{K}} \sum_\ell \rho_\ell k_\ell^2$. Then the maximum number of perfectly isolated tasks is*

$$T_{\max} = \left\lfloor \frac{1}{\bar{\rho}} \right\rfloor.$$

*For all $T \leq T_{\max}$, there exists a disjoint allocation with zero interference and forgetting margin*

$$\epsilon_f \geq c \sum_\ell \|P_{\Omega_t^{(\ell)}}((\boldsymbol{U}_L^{(\ell)})^\top \boldsymbol{G}_{j_t}^{(\ell)} \boldsymbol{U}_R^{(\ell)})\|_F,$$

*for a universal constant $c > 0$.*

When $T > T_{\max}$, disjoint allocation is impossible and supports overlap. Assuming uniform random allocation:

**Theorem 4.3** (Controlled degradation beyond capacity). *For $T > T_{\max}$,*

$$\mathbb{E}[\delta] = O\left(\sum_\ell \frac{\rho_\ell^2 T}{k_\ell^2} \|\boldsymbol{G}^{(\ell)}\|_F \|\Delta \boldsymbol{W}^{(\ell)}\|_F\right), \quad \mathbb{E}[\eta] = O\left(\sum_\ell \frac{\rho_\ell^2 (T-1)}{k_\ell^2} \|\boldsymbol{G}^{(\ell)}\|_F \|\Delta \boldsymbol{W}^{(\ell)}\|_F\right),$$

*and*

$$\mathbb{E}[\epsilon_f] \geq \sum_\ell \left(1 - \frac{\rho_\ell(T-1)}{k_\ell^2}\right) \|P_{\Omega_t^{(\ell)}}((\boldsymbol{U}_L^{(\ell)})^\top \boldsymbol{G}_{j_t}^{(\ell)} \boldsymbol{U}_R^{(\ell)})\|_F.$$

Thus interference grows only linearly with $T$ and quadratically with sparsity $\rho$, much milder than the uncontrolled blow-up of naive approaches.

### 4.3 PROVABLE GUARANTEES

Let $\gamma_\ell = \|P_{\Omega_t^{(\ell)}}((\boldsymbol{U}_L^{(\ell)})^\top \boldsymbol{G}_{j_t}^{(\ell)} \boldsymbol{U}_R^{(\ell)})\|_F$ be the target gradient energy captured.

**Theorem 4.4** (Provable continual unlearning). *There exist constants $c_1, c_2 > 0$ such that:*

- *If $T \leq T_{\max}$, then $\delta = \eta = 0$ and $\Delta L_{j_t} \geq c_1 \sum_\ell \gamma_\ell^2$.*

- *If $T > T_{\max}$, then*

$$\mathbb{E}[\Delta L_{j_t}] \geq c_2 \sum_\ell \left(1 - \frac{\rho_\ell(T-1)}{k_\ell^2}\right) \gamma_\ell^2, \quad \mathbb{E}[\delta], \mathbb{E}[\eta] = O\left(\sum_\ell \frac{\rho_\ell^2 T}{k_\ell^2}\right).$$

Hence spectral decomposition yields exact satisfaction of the three conditions within capacity and controlled degradation beyond it.

### 4.4 UNIFIED INFERENCE AND ROUTING

The same parameterization provides an inference-time routing signal. For input $\boldsymbol{x}$, define the *spectral activation norm* (SAN) for task $t$ as

$$\mathrm{SAN}_t(\boldsymbol{x}) = \|\Delta \boldsymbol{W}_t^{(L)} \boldsymbol{h}^{(L-1)}(\boldsymbol{x})\|_2,$$

optionally normalized by $\|\boldsymbol{h}^{(L-1)}(\boldsymbol{x})\|_2$, where $\boldsymbol{h}^{(L-1)}$ is the last hidden representation. Since $\Delta \boldsymbol{W}_t^{(L)}$ is optimized against domain $j_t$, inputs from $j_t$ yield disproportionately large $\mathrm{SAN}_t$.

**Theorem 4.5** (Spectral separation). *Under disjoint supports,*

$$\mathbb{E}_{\boldsymbol{x} \sim \mathcal{P}_{j_t}}[\mathrm{SAN}_t(\boldsymbol{x})] \gg \mathbb{E}_{\boldsymbol{x} \sim \mathcal{P}_{j_s}}[\mathrm{SAN}_t(\boldsymbol{x})], \quad s \neq t,$$

*and similarly against background distributions $\mathcal{P}^O$.*

At inference, each task learns a threshold $\tau_t$; inputs are routed to $t^* = \arg\max_t \mathrm{SAN}_t(\boldsymbol{x})$ if above $\tau_t$, otherwise to the base model $M_{\boldsymbol{\theta}_0}$. Thus both optimization and inference are seamlessly unified together within a single coherent spectral framework.

**Comparison with orthogonal LoRA.** Prior methods Gao et al. (2025) penalize overlap between low-rank adapters, yielding approximate orthogonality without guarantees or capacity accounting. Our approach is constructive: disjoint coefficient supports yield exact orthogonality, explicit capacity laws, and unified routing, a strictly stronger foundation for continual unlearning.

## 5 EXPERIMENT

### 5.1 EXPERIMENT SETUP

**Dataset.** We employ two datasets: TOFU for evaluating unlearning on fictitious knowledge, and CLINC150 for intent classification. **Fictitious Knowledge Generation.** The TOFU benchmark Maini

Table 1: Performance comparison between **SCOPE** and baselines on TOFU datasets under three unlearning requests. S.U. and D.U. denote the accuracy rate of unlearning on synthetic and domain-specific requests (lower is better). Accuracy of R.D. and R.A. denotes the performance on the retained dataset (higher is better).

| Method | Unlearning Request 1 | | | | Unlearning Request 2 | | | | Unlearning Request 3 | | | |
| | Selective Forgetting | | Utility Preservation | | Selective Forgetting | | Utility Preservation | | Selective Forgetting | | Utility Preservation | |
| | S.U.↓ | D.U.↓ | R.D.↑ | R.A.↑ | S.U.↓ | D.U.↓ | R.D.↑ | R.A.↑ | S.U.↓ | D.U.↓ | R.D.↑ | R.A.↑ |
| --- | --- | --- | --- | --- | --- | --- | --- | --- | --- | --- | --- | --- |
| Base | 85.0±0.0 | 90.0±0.0 | 85.8±0.0 | 89.0±0.0 | 87.3±0.0 | 89.3±0.0 | 85.8±0.0 | 89.0±0.0 | 85.3±0.0 | 90.0±0.0 | 85.8±0.0 | 89.0±0.0 |
| GradAsc | 75.0±0.0 | 85.0±0.0 | 81.0±0.0 | 86.0±0.0 | 17.6±0.2 | 23.1±1.1 | 19.0±0.0 | 0.0±0.0 | 17.1±0.9 | 14.2±2.5 | 19.0±0.0 | 0.0±0.0 |
| GradDif | 78.1±0.0 | 84.0±1.7 | 81.9±1.6 | 86.7±0.6 | 62.5±5.4 | 70.0±8.7 | 70.4±3.7 | 65.7±7.2 | 63.3±10.3 | 75.2±4.5 | 19.0±0.0 | 0.0±0.0 |
| EUL | 84.1±0.2 | 86.3±0.6 | 86.1±1.5 | 86.7±1.5 | 90.0±3.3 | 91.0±3.8 | 85.8±0.5 | 88.0±2.0 | 88.1±0.2 | 83.5±0.5 | 83.4±1.0 | 86.3±1.4 |
| PO | 12.5±0.6 | 13.0±1.3 | 78.4±0.2 | 82.7±0.6 | 59.4±8.2 | 58.2±8.3 | 85.2±1.2 | 83.7±2.8 | 58.4±2.8 | 53.4±2.0 | 81.6±0.8 | 83.2±1.3 |
| NPO | 68.8±3.2 | 75.0±0.0 | 83.6±0.4 | 89.0±0.1 | 76.3±8.2 | 84.3±2.3 | 82.1±2.2 | 87.6±0.6 | 77.7±6.7 | 79.2±1.3 | 81.4±0.8 | 87.3±0.7 |
| SOGD | 25.4±0.1 | 76.0±1.7 | 83.0±0.7 | 88.3±0.2 | 22.0±6.9 | 24.0±3.2 | 79.0±3.1 | 83.2±1.6 | 17.0±4.0 | 21.7±6.4 | 80.3±2.0 | 87.6±0.8 |
| SOPO | 25.6±1.0 | 38.0±0.9 | 83.7±0.6 | 85.3±1.2 | 31.4±7.3 | 37.5±2.6 | 85.1±0.5 | 87.3±0.7 | 34.0±3.5 | 40.3±0.5 | 82.2±0.8 | 86.2±0.4 |
| $O^3$ | 12.5±0.5 | 14.4±0.5 | 85.1±0.1 | 89.0±0.0 | 15.8±0.3 | 20.3±0.8 | 85.0±0.0 | 89.0±0.0 | 15.5±0.5 | 19.7±0.7 | 84.9±0.2 | 88.8±0.2 |
| LoKU | 15.5±1.0 | 13.5±0.3 | 82.1±0.4 | 88.4±0.3 | 14.8±0.2 | 19.9±2.2 | 82.1±0.1 | 88.0±0.0 | 15.0±0.2 | 23.7±0.7 | 79.9±0.2 | 87.2±0.2 |
| **SCOPE** | 11.9±0.6 | 14.8±1.2 | 85.3±0.3 | 89.0±0.1 | 16.0±0.2 | 19.1±0.6 | 85.3±0.5 | 89.0±0.0 | 16.5±0.3 | 19.1±0.7 | 84.9±0.2 | 89.0±0.3 |

et al. (2024) contains GPT-4 generated questions about fictitious authors. It defines three forget-sets (forget01/05/10) with 1%, 5%, and 10% of randomly chosen authors as continual unlearning requests, plus 400 retained samples for evaluation. TOFU also includes Real-world Authors and World Facts subsets to assess knowledge preservation. **Intent Classification.** The CLINC150 corpus mis (2020) spans 150 intent classes across five domains, with 200/40/60 samples for train/validation/test per class. We select three privacy-related domains (*work*, *travel*, *home*) as unlearning requests. For utility preservation, MRPC Dolan & Brockett (2005) and RTE Wang et al. are used, focusing on paraphrase detection and textual entailment.

**Evaluation Metrics.** To evaluate the *unlearning effectiveness*, we report accuracy on the unlearning train set and test set, denoted as Sample-level Unlearning (S.U.) and Distribution-level Unlearning (D.U.). To measure *utility preservation*, we comprehensively assess performance on the Retained Distribution (R.D.) as well as several auxiliary benchmarks, including Real Authors (R.A.), World Facts (W.F.), MRPC, and RTE. Finally, to evaluate the *detection capability*, we adopt the widely used Area Under the ROC Curve (AUROC) on OOD detection tasks. Lower values of S.U. and D.U. indicate stronger unlearning, while higher values of R.D., R.A., W.F., MRPC, RTE, and AUROC consistently reflect better knowledge preservation and detection ability.

**Implementation Details.** Following TOFU Maini et al. (2024) and SOPO Jia et al. (2024), we use LLaMA2-7b Touvron et al. (2023) as the target model. All experiments are repeated with three random seeds. We adopt the size of the sparse coefficient matrix $\|\Omega_t\| = 70000$ for one unlearning request $t$, with FourierFT scale set to 300 and a batch size of 128 for the combined datasets.

**Baseline.** To better demonstrate the effectiveness of our proposed methods, we implement a series of state-of-the-art language model unlearning approaches: GradAscGolatkar et al. (2020), GradDifYao et al. (2023), EULChen & Yang (2023), POEldan & Russinovich (2023), NPOZhang et al. (2024), SOGDJia et al. (2024), SOPOJia et al. (2024), and $O^3$Gao et al. (2025) LoKUCao & Yang (2015) We only conduct reasonable modifications to customize them in our continual unlearning settings.

## 5.2 RESULTS

**TOFU dataset.** Table 1 presents the results on TOFU across three unlearning requests. **Selective Forgetting.** Our method achieves low S.U. and D.U. in all settings, showing stronger forgetting of fictitious knowledge. For instance, in Unlearning Request 1, it reduces S.U. and D.U. to 11.9 and 14.8, outperforming $O^3$ (12.5/14.4). Even under Unlearning Request 3, where GradAsc and GradDif seem competitive on unlearning metrics, they collapse on utility (R.D. ≈19.0, R.A. = 0.0), failing to generalize beyond the forget set. **Utility Preservation.** Our method maintains high R.D. (≈85) and R.A. (≈89) across requests, in contrast to baselines like PO that sacrifice retained knowledge. This highlights our framework's ability to balance effective unlearning with minimal interference on real authors and world facts. **Unlearning Persistence.** Across multiple requests, once knowledge is

Table 2: Performance comparison between **SCOPE** and baselines on CLINC150 intent classification under three unlearning requests.

| Metric | Base | GradDif | EUL | PO | NPO | SOGD | SOPO | $O^3$ | LoKU | **SCOPE** |
|--------|------|---------|-----|----|----|------|------|-----|------|-------|
| **Unlearning Request 1** | | | | | | | | | | |
| S.U.↓ | 100.0±0 | 0.1±0.2 | 0.1±0.2 | 26.3±15.1 | 99.9±0.1 | 0±0 | 24.9±15.6 | 10.3±8.1 | 99.5±0.1 | 9.2±1.2 |
| D.U.↓ | 99.9±0 | 0±0 | 0±0 | 26.7±14.0 | 99.0±0 | 0±0 | 26.3±15.0 | 14.3±0.3 | 99.0±0.0 | 10.3±0.3 |
| R.D.↑ | 99.8±0 | 90.8±3.4 | 98.3±0.4 | 99.3±0.3 | 99.2±0.2 | 92.3±0.9 | 99.6±0.1 | 98.9±0.1 | 98.3±0.2 | 99.3±0.4 |
| MRPC↑ | 88.0±0 | 39.9±3.4 | 87.2±0.1 | 84.1±0.2 | 87.3±0.3 | 6.1±3.6 | 85.5±0.6 | 84.8±0.1 | 87.0±0.3 | 86.2±1.2 |
| RTE↑ | 88.7±0 | 31.6±5.3 | 88.1±0 | 86.3±1.1 | 88.4±0.4 | 17.9±6.4 | 87.1±1.1 | 87.5±0.6 | 87.1±0.3 | 88.1±0.4 |
| **Unlearning Request 2** | | | | | | | | | | |
| S.U.↓ | 100.0±0 | 0±0 | 0.1±0.2 | 59.6±3.0 | 99.9±0.1 | 0±0 | 62.3±1.4 | 50.5±0.8 | 99.9±0.1 | 49.1±1.1 |
| D.U.↓ | 99.9±0 | 0±0 | 0±0 | 59.8±3.0 | 99.3±0.3 | 0.1±0.1 | 60.3±1.9 | 55.6±0.6 | 99.2±0.2 | 52.2±0.5 |
| R.D.↑ | 99.8±0 | 12.7±3.6 | 87.6±3.3 | 99.4±0.2 | 99.2±0.3 | 93.1±2.0 | 99.6±0.2 | 94.1±0.8 | 99.1±0.2 | 98.9±1.3 |
| MRPC↑ | 88.0±0 | 9.0±3.8 | 80.3±3.1 | 87.3±0.1 | 87.2±0.7 | 3.3±3.2 | 87.1±0.2 | 87.0±0.2 | 87.1±0.6 | 87.3±0.4 |
| RTE↑ | 88.7±0 | 0.8±0.8 | 82.9±3.1 | 88.0±0.2 | 88.9±0.6 | 19.5±9.0 | 87.7±1.1 | 89.3±0.2 | 88.8±0.5 | 88.9±0.2 |
| **Unlearning Request 3** | | | | | | | | | | |
| S.U.↓ | 99.9±0 | 0±0 | 0±0 | 56.2±5.4 | 99.9±0.1 | 0±0 | 58.8±15.5 | 40.6±4.0 | 99.9±0.1 | 35.6±4.3 |
| D.U.↓ | 99.9±0 | 0±0 | 0±0 | 56.7±4.8 | 99.2±0.2 | 0.1±0.1 | 59.7±14.8 | 42.4±3.8 | 99.3±0.1 | 44.3±5.1 |
| R.D.↑ | 99.8±0 | 75.5±4.8 | 92.3±5.2 | 99.0±0.4 | 99.3±0.1 | 94.0±1.8 | 99.6±0.3 | 97.8±0.8 | 99.2±0.2 | 99.7±0.1 |
| MRPC↑ | 88.0±0 | 12.9±6.0 | 81.3±2.1 | 86.3±0.2 | 87.0±0.4 | 2.9±0.6 | 86.6±1.2 | 86.6±1.0 | 87.2±0.3 | 87.7±0.4 |
| RTE↑ | 88.7±0 | 1.7±2.1 | 76.3±4.0 | 87.0±0.4 | 88.9±0.2 | 23.7±9.9 | 86.0±1.4 | 89.0±0.2 | 88.7±0.4 | 88.9±0.4 |

Table 3: $\text{SAN}_t$(OOD detection mechanism) performance comparision between **SCOPE** and other baselines on TOFU (AUROC, %).

| Task | Fictitious Knowledge Generation | | | | | | | | |
|------|------|------|------|------|------|------|------|------|------|
| | TOFU-forget01 | | | TOFU-forget05 | | | TOFU-forget10 | | |
| ID/OOD | R.D. | R.A. | W.F. | R.D. | R.A. | W.F. | R.D. | R.A. | W.F. |
| MDF | 90.5 | 96.6 | 97.6 | 80.3 | 92.7 | 98.3 | 91.3 | 97.8 | 98.8 |
| Agg | 94.4 | 98.0 | 98.0 | 81.9 | 94.0 | 98.5 | 85.0 | 97.5 | 99.0 |
| **SCOPE** w/o $\text{SAN}_t$ | 90.2 | 93.2 | 95.3 | 75.2 | 72.2 | 81.2 | 83.5 | 88.5 | 95.4 |
| **SCOPE** | 95.5 | 98.5 | 98.0 | 87.6 | 95.0 | 99.0 | 87.9 | 98.8 | 99.1 |

forgotten, it does not resurface. For example, our S.U. and D.U. remain stable, whereas methods like SOPO fluctuate, indicating partial recovery of forgotten knowledge.

**CLINIC 150 dataset** Table 2 presents the results on CLINC150 when continually unlearning the domains *work*, *travel*, and *home*. **Selective Forgetting.** Our method achieves the lowest S.U. and D.U. in most cases, thereby clearly demonstrating strong removal of domain-specific intents. For instance, in Unlearning Request 1, our approach reduces S.U. and D.U. to 9.2 and 10.3, while competing methods either fail to unlearn (LoKU, NPO) or collapse (GradDif, SOGD). **Utility Preservation.** At the same time, our method reliably preserves R.D. at 99.3 and maintains MRPC and RTE accuracies at 86.2 and 88.1, comparable to or better than baselines. Under Unlearning Request 2 and Request 3, our framework again achieves significantly lower unlearning errors (49.1/52.2 and 35.6/44.3) than $O^3$ (50.5/55.6 and 40.6/42.4), while sustaining high retained accuracy. **Unlearning Persistence.** As the number of unlearning requests increases, our method consistently sustains degradation on the forgotten domains without rebound. By contrast, some baselines exhibit "forget–relearn" oscillations, where forgotten intents partially reappear after subsequent training. Our framework consistently enforces long-term forgetting across sequential requests.

## 5.3 ABLATION STUDY

**OOD detection mechanism.** Table 3 presents the OOD detection results on the Fictitious Knowledge Generation task. Our method consistently outperforms baselines MDF Xu et al. (2021) and Agg Darrin et al. (2024) across all unlearning requests. On TOFU-forget01, our detector achieves 95.5% AUROC on R.D., surpassing MDF (90.5%) and Agg (94.4%). Our method demonstrates superior robustness as task complexity increases.

Figure 2: Performance comparison on TOFU-forget10 under 10 continual unlearning requests.

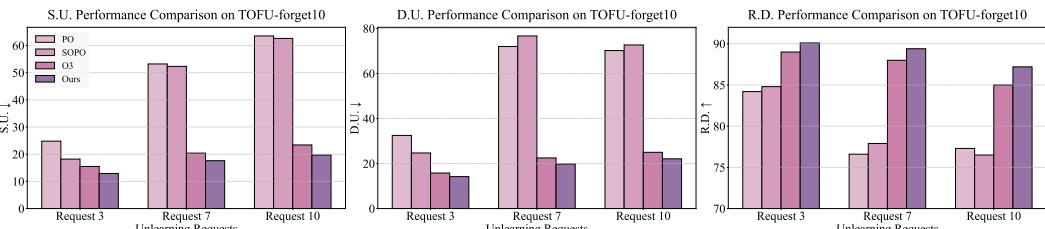

In challenging scenarios like TOFU-forget02 and TOFU-forget10, our approach sustains strong performance while baselines degrade significantly. This widening performance gap with increasing unlearning requests highlights our method's reliable generalization capability. We also maintain the highest scores on R.A. and W.F. metrics consistently. These results provide empirical evidence that spectral update magnitude serves as a more principled and effective indicator of task relevance compared to direct reliance on textual inputs.

Table 4: Comparison of training cost across **SCOPE** and baselines.

| Method | OOD Train Param. | Model Train Param. |
|---|---|---|
| Baseline w/o $\mathcal{O}_3$ | 355M | 6,758M |
| $\mathcal{O}_3$ | 355M | 20M |
| **SCOPE** | 56M | 11M |

**Scale of unleqarning requests.** Figure 2 divides these 20 fictional authors evenly into 10 groups on TOFU-forget10, resulting in 10 unlearning requests. Each request adds information about 2 additional fictional authors based on the previous request. **Selective Forgetting**: Based on the S.U. metric performance, our method demonstrates exceptional capability in selective forgetting. The D.U. metric reflects the model's ability to maintain its original functionality after forgetting specific information. Our method achieves optimal utility preservation across all requests (14.2, 19.8, 22.1), significantly outperforming baseline methods. This indicates that our method can precisely identify and forget target information (fictional author knowledge).

**Utility Preservation.** The R.D. metric proves the accuracy of forgetting from another dimension. Although all methods show declining R.D. scores as the complexity of unlearning tasks increases, our method exhibits relatively smaller decline and maintains high performance levels.

**Scale of sparse coefficient matrix.** Table 5 shows that as sparse coefficient matrix $\|\Omega_t\|$ increases, both S.U. and D.U. consistently decrease, while R.D. and R.A. steadily improve.

Table 5: Performance across scale of the sparse coefficient matrix $\|\Omega_t\| = 70000$ on TOFU under unlearning request 1.

| $n$ | S.U.$\downarrow$ | D.U.$\downarrow$ | R.D.$\uparrow$ | R.A.$\uparrow$ |
|---|---|---|---|---|
| 20,000 | 18.5 | 17.6 | 80.4 | 86.5 |
| 50,000 | 13.5 | 17.6 | 83.0 | 87.8 |
| 70,000 | 11.9 | 14.8 | 85.3 | 89.0 |
| 100,000 | **10.7** | **12.2** | **86.6** | **90.1** |

For example, increasing $\|\Omega_t\|$ from 20,000 to 100,000 reduces S.U. from 18.5 to 10.7 and D.U. from 17.6 to 12.2, while R.D. improves from 80.4 to 86.6 and R.A. rises from 86.5 to 90.1. This demonstrates that larger $\|\Omega_t\|$ values not only enable more effective unlearning but also preserve higher utility, validating the benefit of frequency-conditioned adapters.

**Training Efficiency Analysis.** Our method achieves exceptional parameter efficiency, requiring only 56M OOD parameters and 11.2M model parameters, which correspond to a 84.2% reduction and a 99.8% reduction, respectively, compared to the baseline that trains over 355M and 6.8B parameters. While $\mathcal{O}_3$ reduces model parameters to 20M, it still requires 355M OOD parameters, making our approach 96% more efficient overall (16.2M vs. 375M total parameters).

## 6 CONCLUSION

We introduced a principled framework for continual unlearning in large language models, deriving explicit conditions for forgetting, preservation, and persistence, and showing that these can be satisfied exactly through a spectral parameterization. This provides the first capacity-aware theoretical guarantees for sequential unlearning, while also enabling a unified mechanism for inference-time routing. Our experiments validate that these guarantees translate into practical gains in utility preservation, persistence, and efficiency.

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

# A PROOFS

We now provide proofs for the main theorems in Section 4. For clarity, we first state the assumptions under which our analysis holds.

## A.1 ASSUMPTIONS

- **Smoothness.** Each domain loss $L_i(\boldsymbol{\theta})$ is differentiable with $L$-Lipschitz continuous gradients. That is, for all $\boldsymbol{\theta}, \boldsymbol{\theta}'$,

$$\|g_i(\boldsymbol{\theta}) - g_i(\boldsymbol{\theta}')\|_2 \leq L\|\boldsymbol{\theta} - \boldsymbol{\theta}'\|_2.$$

  This ensures the validity of the first-order approximation in Section 3.2.
- **Small-step updates.** Each update $\Delta\boldsymbol{\theta}_t$ satisfies $\|\Delta\boldsymbol{\theta}_t\|_2 \ll \|\boldsymbol{\theta}_{t-1}\|_2$, so higher-order Taylor terms can be absorbed into tolerances $(\epsilon_t, \delta, \eta)$.
- **Spectral bases.** For each layer $\ell$, the matrices $\boldsymbol{U}_L^{(\ell)}$ and $\boldsymbol{U}_R^{(\ell)}$ are orthonormal. Our results hold for any orthonormal basis; FFT is used in practice for computational efficiency.
- **Random allocation beyond capacity.** When $T > T_{\max}$, coefficient supports are assumed to be assigned uniformly at random without replacement. This enables expectation-based bounds on degradation.

These mild assumptions are standard in optimization theory and continual learning analysis, and they align with prior work on orthogonal adapters and spectral parameterizations. We now proceed with the step-by-step proofs.

## A.2 PROOF OF THEOREM 4.1

*Proof.* **Step 1: Expand the inner product.** For any preserved or previously forgotten domain $q$,

$$\langle g_q(\boldsymbol{\theta}_{t-1}), \Delta\boldsymbol{\theta}_t \rangle = \sum_{\ell=1}^{L} \langle \boldsymbol{G}_q^{(\ell)}, \Delta\boldsymbol{W}_t^{(\ell)} \rangle_F.$$

By spectral parameterization, $\Delta\boldsymbol{W}_t^{(\ell)} = \boldsymbol{U}_L^{(\ell)} \boldsymbol{S}_t^{(\ell)} (\boldsymbol{U}_R^{(\ell)})^\top$.

**Step 2: Basis factorization.**

$$\langle \boldsymbol{G}_q^{(\ell)}, \Delta\boldsymbol{W}_t^{(\ell)} \rangle_F = \langle (\boldsymbol{U}_L^{(\ell)})^\top \boldsymbol{G}_q^{(\ell)} \boldsymbol{U}_R^{(\ell)}, \ \boldsymbol{S}_t^{(\ell)} \rangle_F.$$

**Step 3: Disjoint support implies orthogonality.** If $\Omega_t^{(\ell)} \cap \Omega_q^{(\ell)} = \emptyset$, the inner product vanishes for every $\ell$.

**Step 4: Summation across layers.** Thus $\langle g_q(\boldsymbol{\theta}_{t-1}), \Delta\boldsymbol{\theta}_t \rangle = 0$, so preservation and persistence hold exactly. Forgetting holds whenever the target gradient overlaps with $\Omega_t^{(\ell)}$. $\qquad\square$

## A.3 PROOF OF THEOREM 4.2

*Proof.* **Step 1: Count available coefficients.** Each layer $\ell$ provides $k_\ell^2$ spectral coefficients, so the budget is $\mathcal{K} = \sum_\ell k_\ell^2$.

**Step 2: Per-task allocation.** Task $t$ uses $\sum_\ell \rho_\ell k_\ell^2$. Let $\bar{\rho} = \frac{1}{\mathcal{K}} \sum_\ell \rho_\ell k_\ell^2$.

**Step 3: Maximal disjoint allocation.** The number of perfectly isolated tasks is

$$T_{\max} = \left\lfloor \frac{1}{\bar{\rho}} \right\rfloor.$$

**Step 4: Forgetting margin.** With disjoint allocation,

$$\epsilon_f \geq c \sum_\ell \|P_{\Omega_t^{(\ell)}}((\boldsymbol{U}_L^{(\ell)})^\top \boldsymbol{G}_{j_t}^{(\ell)} \boldsymbol{U}_R^{(\ell)})\|_F.$$

$\qquad\square$

### A.4 PROOF OF THEOREM 4.3

*Proof.* **Step 1: Overlap probability.** When $T > T_{\max}$, supports overlap with probability $\rho_\ell(T - 1)/k_\ell^2$.

**Step 2: Preservation violation.** Expected drift is

$$\mathbb{E}[\delta] = O\left(\sum_\ell \frac{\rho_\ell^2 T}{k_\ell^2} \|\boldsymbol{G}^{(\ell)}\|_F \|\Delta \boldsymbol{W}^{(\ell)}\|_F\right).$$

**Step 3: Persistence violation.** Similarly,

$$\mathbb{E}[\eta] = O\left(\sum_\ell \frac{\rho_\ell^2 (T - 1)}{k_\ell^2} \|\boldsymbol{G}^{(\ell)}\|_F \|\Delta \boldsymbol{W}^{(\ell)}\|_F\right).$$

**Step 4: Forgetting margin.** Effective energy is reduced by overlaps:

$$\mathbb{E}[\epsilon_f] \geq \sum_\ell \left(1 - \frac{\rho_\ell(T-1)}{k_\ell^2}\right) \|P_{\Omega_t^{(\ell)}}((\boldsymbol{U}_L^{(\ell)})^\top \boldsymbol{G}_{j_t}^{(\ell)} \boldsymbol{U}_R^{(\ell)})\|_F.$$

$\square$

### A.5 PROOF OF THEOREM 4.4

*Proof.* **Step 1: Within-capacity case.** If $T \leq T_{\max}$, Theorem 4.2 ensures disjoint allocation. Thus $\delta = \eta = 0$ and $\Delta L_{j_t} \geq c_1 \sum_\ell \gamma_\ell^2$.

**Step 2: Beyond-capacity case.** If $T > T_{\max}$, Theorem 4.3 gives

$$\mathbb{E}[\Delta L_{j_t}] \geq c_2 \sum_\ell \left(1 - \frac{\rho_\ell(T-1)}{k_\ell^2}\right)\gamma_\ell^2,$$

with $\mathbb{E}[\delta], \mathbb{E}[\eta] = O\left(\sum_\ell \frac{\rho_\ell^2 T}{k_\ell^2}\right)$.

**Step 3: Combine.** Together, these establish Theorem 4.4. $\square$

### A.6 PROOF OF PROPOSITION 3.5

*Proof.* **Step 1: First-order expansion.** From the Taylor approximation, for any domain $i$,

$$L_i(\boldsymbol{\theta}_t) \approx L_i(\boldsymbol{\theta}_{t-1}) + \langle g_i(\boldsymbol{\theta}_{t-1}), \Delta\boldsymbol{\theta}_t\rangle.$$

**Step 2: Apply condition definitions.**

- *Selective forgetting:* $L_{j_t}(\boldsymbol{\theta}_t) \geq L_{j_t}(\boldsymbol{\theta}_{t-1}) + \epsilon_t$ implies $\langle g_{j_t}(\boldsymbol{\theta}_{t-1}), \Delta\boldsymbol{\theta}_t\rangle \geq \epsilon_t$.

- *Utility preservation:* $|L_i(\boldsymbol{\theta}_t) - L_i(\boldsymbol{\theta}_{t-1})| \leq \delta$ for $i \notin \{j_1, \ldots, j_t\}$ implies $|\langle g_i(\boldsymbol{\theta}_{t-1}), \Delta\boldsymbol{\theta}_t\rangle| \leq \delta$.

- *Persistence:* $L_{j_s}(\boldsymbol{\theta}_t) \geq L_{j_s}(\boldsymbol{\theta}_s) - \eta$ for $s < t$ implies $|\langle g_{j_s}(\boldsymbol{\theta}_{t-1}), \Delta\boldsymbol{\theta}_t\rangle| \leq \eta$.

**Step 3: Combine.** Each of the three conditions translates directly to the stated inner product inequalities, proving the proposition. $\square$

## B THE USE OF LARGE LANGUAGE MODELS (LLMS)

After completing the initial draft, we used LLMs to polish and refine the writing. They edit our typos, and improve consistency of style across the paper. All technical content and results remain fully authored and verified by us; the LLMs served only as writing assistants.

