# OpenReview forum: "Provably Continual Unlearning for Large Language Model"
_ICLR.cc/2026/Conference — ICLR 2026 Conference Withdrawn Submission_

### Official Review · Reviewer_Lien · 2025-10-28

**Soundness:** 3
**Presentation:** 2
**Contribution:** 2
**Rating:** 2
**Confidence:** 3

**Summary:**

This work studies continual unlearning, in which unlearning requests of data arrive sequentially rather than as a one-time thing. The authors propose SCOPE as a framework in which this can be accomplished under three desiderata: selective forgetting, utility preservation, and persistence. Empirical results are presented to support their argument.

**Strengths:**

- The formalization into three specific conditions for sequential unlearning is useful as a starting point.
- The FFT-based representation of $U$ is a neat idea, and I believe it could have some potential for unlearning methods in practice.
- Experiments are provided to back up the results.

**Weaknesses:**

- The theory provided for the unlearning guarantees are quite simplistic, and the proofs in the appendix could be way less verbose (and possibly even combined in the main text). On the other hand, I believe more discussion could be given on the background for FFT, as well as motivation for how to select the size of the Fourier bases.
- For the experiments, it seems that SCOPE does significantly better than other baselines, which is a bit shocking to me. See questions below.
- Many citations are formatted slightly incorrectly (e.g. include parenthesis when using a citation in a sentence)

**Questions:**

- Regarding the experiments: is there something fundamental about your unlearning algorithm via FFT that allows such a big improvement over other methods? My understanding as that design here is mainly to allow for greater efficiency in the storage of the data representation.
- The given desiderata are quite similar to existing literature (e.g. [1]); how does your approach perform with respect to (for instance, privacy leakage)?

[1] Shi et al. (2024), MUSE: Machine Unlearning Six-Way Evaluation for Language Models

---

### Official Review · Reviewer_1zBW · 2025-10-29

**Soundness:** 2
**Presentation:** 2
**Contribution:** 2
**Rating:** 4
**Confidence:** 1

**Summary:**

This paper proposes SCOPE, a spectral framework for continual unlearning in LLMs, using orthonormal bases like FFT to enforce orthogonality and provide guarantees on forgetting, preservation, and persistence without retained data.

**Strengths:**

The approach innovatively uses spectral decomposition to achieve provable orthogonality by design, offering capacity bounds and efficient FFT implementation, which could scale well for sequential unlearning requests.

**Weaknesses:**

The theoretical guarantees rely on strong assumptions like first-order approximations and disjoint supports, which may not hold in practice for complex LLMs, potentially leading to uncontrolled interference beyond capacity.

**Questions:**

How does the method perform on real-world LLMs with overlapping domains? Are there empirical validations beyond TOFU and CLINC150? What is the computational overhead for large models?

---

### Official Review · Reviewer_BY4T · 2025-10-29

**Soundness:** 3
**Presentation:** 2
**Contribution:** 2
**Rating:** 4
**Confidence:** 3

**Summary:**

The authors propose **Spectral Orthogonality for Continual Unlearning with Provable Guarantees (SCOPE)**, a framework that formalizes continual unlearning. They specify three conditions: *selective forgetting* (the requested knowledge is forgotten), *utility preservation* (performance on other domains is maintained), and *persistence* (previously forgotten domains remain forgotten after later updates). They propose a method that satisfies these conditions by parameterizing updates in an orthonormal spectral basis with disjoint coefficient supports. They provide capacity and interference bounds, and later discuss an inference-time setting.

The authors evaluate the method on two datasets: TOFU (fictitious authors) and CLINC150 (intent classification), using LLaMA-2-7B, demonstrating effective unlearning and utility preservation. They also present an OOD detection mechanism. Additionally, the method is evaluated under different unlearning request scales and sparsity levels.

**Strengths:**

The paper presents an original and principled framework for continual unlearning with provable guarantees.

---

Strengths:
- The introduction is clear and uses helpful examples.
- The problem is formulated nicely and is easy to follow.
- The paper discusses complexity.

**Weaknesses:**

The draft would benefit from clearer explanations, more consistent presentation, and additional detail to improve readability and understanding.

---

### **Abstract & Writing Clarity**

- The abstract is too condensed, which makes it difficult to understand.
- The sentence beginning on page 1 L22 is overly complex and hard to follow.
- There are several typos and minor grammatical issues throughout; the text would benefit from polishing.
- Section 3 would benefit from additional intuition to help readers understand the key ideas.
- The statement on page 3 L117 reads as an unsupported opinion rather than a fact.

---

### **Definitions & Notation**

- The concept of inference-time routing is mentioned but never properly explained.
- Number of layers and loss use the same symbol.
- The use of subscripts (i) and (t) is confusing; their meaning and relationship should be clarified.
- The definition of retained samples is unclear.
- The persistence metric is mentioned but skipped in the metrics definition.
- The OOD mechanism is not clearly explained.

---

### **Related Work**

- The related work section feels more like a general introduction or background rather than a focused discussion of relevant research. It should more clearly connect prior methods to this work.

---

### **Figures, Tables, and Formatting**

- The figures contain text that is too small to read comfortably.
- Table 1 and Table 2 have different format (flipped); consistent formatting should be used.
- Table 4 is never referenced or discussed in the text.

---

### **Methodology & Technical Presentation**

- The paper repeatedly mentions FFT but never explains why it is used in depth.
- The baseline methods are not described in enough detail to understand the comparisons.
- It is unclear what the second dataset is supposed to represent.
- It is unclear where forget-relearn oscillations can be seen in the results.
- The meaning of 'collapse of scores' is not defined, nor is it explained why a value of 0 is considered bad for selective foregetting.

**Questions:**

1. Can you explain the different usage of subscripts (i) and (t), is (i) the knowledge subscript and (t) the current task, how are they related?
2. *“For example, our S.U. and D.U. remain stable, whereas methods like SOPO fluctuate, indicating partial recovery of forgotten knowledge”*,  which metric shows that fluctuation?

---

### Official Review · Reviewer_QV9a · 2025-11-01

**Soundness:** 2
**Presentation:** 2
**Contribution:** 2
**Rating:** 4
**Confidence:** 3

**Summary:**

This paper introduces SCOPE, a theoretical and algorithmic framework for continual unlearning in LLM. The authors formalize continual unlearning as satisfying three conditions — selective forgetting, utility preservation, and persistence — and derive a constrained optimization formulation. To meet these conditions, they propose a spectral decomposition of parameter updates with disjoint coefficient supports, ensuring orthogonality and bounding interference between sequential unlearning tasks. The method offers provable guarantees for interference control and scalability via FFT-based implementation, enabling efficient unlearning without stored bases. Experiments on TOFU and CLINC150 datasets show improved unlearning–utility trade-offs compared to baselines such as O3.

**Strengths:**

1. The paper is among the first to define continual unlearning with explicit mathematical conditions (forgetting, preservation, persistence) and analyze the corresponding optimization landscape.
2. The introduction of Spectral Activation Norm (SAN) for routing is an interesting bridge between optimization and inference.
3. The experiments on both synthetic (TOFU) and real-world (CLINC150) benchmarks demonstrate consistent performance, particularly in preserving retained domain utility.

**Weaknesses:**

1. The main concern is that evaluation lacks scale and diversity. Experiments are limited to small-scale benchmarks (TOFU, CLINC150) and LLaMA2-7B; no evidence is shown on larger or instruction-tuned models where continual interference is more realistic.
2. While the spectral basis formulation is elegant, the core mechanism (orthogonal subspace allocation) is conceptually similar to prior orthogonal LoRA and O3 approaches, with the “provable” aspect largely built upon strong simplifying assumptions.
3. Theoretical guarantees may not hold in practice. The proofs rely on idealized assumptions, which are unlikely to strictly hold in deep LLM optimization.
4. Practical gains marginal over strong baselines. Improvements over O3 and SOPO are small (≈1–2% in unlearning metrics), which may not justify the complexity of the spectral parameterization.
5. No ablation on key theoretical design choices. The paper lacks analysis of the impact of basis choice (FFT vs. random orthogonal), sparsity allocation, or how the spectral capacity bound relates to model layers empirically.

**Questions:**

1. How sensitive is SCOPE to the choice of spectral basis (e.g., FFT vs. random vs. learned bases)?
2. Does the Spectral Activation Norm generalize to real-world OOD inputs beyond synthetic domains like TOFU?
3. The results are only on LLaMA2-7B. How would this scale to more recent large models (e.g., LLaMA3-8B, Mistral-Instruct, Qwen-series)?
4. In the introduction, the authors mention toxic content as an example of unlearning targets, but the paper never experiments with or defines a harmful-data continual unlearning setting (WMDP). How would the method handle concept-level or toxicity-based unlearning tasks?
5. Some citations appear inconsistently formatted, especially in section 5.1 where author names and years are merged or misplaced (e.g., *GradAscGolatkar et al. (2020)*). These should follow standard academic style.

---

### Note · Authors · 2026-01-18

I have read and agree with the venue's withdrawal policy on behalf of myself and my co-authors.